# Maternal Vitamin D Deficiency Impairs the Development of β Cells in Offspring Rats in a Sex-Dependent Manner

**DOI:** 10.3390/ijms25084136

**Published:** 2024-04-09

**Authors:** Aline Z. Schavinski, Natany G. Reis, Henrique J. N. Morgan, Ana Paula Assis, Matheus L. Moro, Rafael R. Valentim, Ana Carolina Seni-Silva, Ester S. Ramos, Isis C. Kettelhut, Luiz C. C. Navegantes

**Affiliations:** 1Department of Physiology, Ribeirão Preto Medical School, University of São Paulo, Ribeirão Preto 14049-900, SP, Brazil; alinezanatta@usp.br (A.Z.S.); natany.garcia.reis@gmail.com (N.G.R.); morgan.hjn@usp.br (H.J.N.M.); matheusmoro@usp.br (M.L.M.); rafaelrossiphd@gmail.com (R.R.V.); 2Department of Biochemistry/Immunology, Ribeirão Preto Medical School, University of São Paulo, Ribeirão Preto 14049-900, SP, Brazil; assis.anap92@gmail.com (A.P.A.); idckette@fmrp.usp.br (I.C.K.); 3Department of Genetics, Ribeirão Preto Medical School, University of São Paulo, Ribeirão Preto 14049-900, SP, Brazil; ana.seni@usp.br (A.C.S.-S.); esramos@usp.br (E.S.R.)

**Keywords:** vitamin D, fetal programming, β cell function, sexual dimorphism

## Abstract

Recent studies have shown that maternal vitamin D deficiency (VDD) causes long-term metabolic changes in offspring. However, little is known about the impact of maternal VDD on offspring endocrine pancreas development and insulin secretion in the adult life of male and female animals. Female rats (Wistar Hannover) were fed either control (1000 IU Vitamin D3/kg), VDD (0 IU Vitamin D3/kg), or a Ca^2+^-enriched VDD diet (0 IU Vitamin D3/kg + Ca^2+^ and P/kg) for 6 weeks and during gestation and lactation. At weaning, VDD status was confirmed based on low serum calcidiol levels in dams and pups. Next, male and female offspring were randomly separated and fed a standard diet for up to 90 days. At this age, serum calcidiol levels were restored to normal levels in all groups, but serum insulin levels were decreased in VDD males without affecting glucagon levels, glycemia, or glucose tolerance. Islets isolated from VDD males showed lower insulin secretion in response to different glucose concentrations, but this effect was not observed in VDD females. Furthermore, VDD males, but not females, showed a smaller total pancreatic islet area and lower β cell mass, an effect that was accompanied by reduced gene expression of *Ins1*, *Ins2*, *Pdx1*, and *SLC2A2*. The decrease in *Pdx1* expression was not related to the methylation profile of the promoter region of this gene. Most of these effects were observed in the male VDD+Ca^2+^ group, indicating that the effects were not due to alterations in Ca^2+^ metabolism. These data show that maternal VDD selectively impairs the morphology and function of β cells in adult male offspring rats and that female offspring are fully protected from these deleterious effects.

## 1. Introduction

It is well established that maternal nutritional status along with environmental or physical conditions have major impacts on intrauterine growth and, consequently, fetal development [1]. The term “fetal programming” was first defined by the epidemiologist David Barker and states that “maternal nutrition that the fetus is exposed to during pregnancy and after birth permanently programs its structure and metabolism, determining susceptibility to chronic diseases later in life” [2]. For unknown reasons, female offspring seem to be more protected from these insults than male offspring. It is known that in models of protein restriction during pregnancy, female offspring respond differently from males, in many cases being protected from insults that occur in β cells, altering insulin secretion and glucose homeostasis in male adulthood [3].

One of the important nutrients that contributes to regulating the cell cycle, tissue development, and functionality is vitamin D or calcitriol. Although the relevance and definition of vitamin D deficiency (VDD) are still under debate, several studies indicate that VDD during pregnancy is highly prevalent [4] and has been linked to a number of long-term risks in offspring, including obesity [5], cardiovascular disease [6], insulin resistance [7], and diabetes mellitus [8].

The majority of vitamin D is obtained from the skin by Ultraviolet-B-induced conversion of 7-dehydrocholesterol to cholecalciferol (Vitamin D3, animal origin) followed by two hydroxylations. The first hydroxylation occurs in the liver by the enzyme 25-hydroxylase (CYP2R1), giving rise to calcidiol (25-hydroxyvitamin D). Calcidiol is transported to the kidney, where it undergoes a second hydroxylation by 1-α hydroxylase (CYP27B1), forming calcitriol (1,25 dihydroxyvitamin D), the active form of vitamin D [9]. The biological actions of vitamin D are mediated by its coupling to the vitamin D receptor (VDR) followed by a conformational change that allows for heterodimerization with the retinoid X receptor (RXR). The VDR/RXR complex binds to vitamin D responsive elements (VDREs) in the promoter region of hundreds of genes, thus regulating their transcription and protein biosynthesis [9].

VDR has been found to be expressed in a variety of cells and organs including pancreatic islets [9,10,11]. In β cells, vitamin D directly acts by binding to VDR or by local production from the main circulating form, calcidiol [10]. Indeed, Bland et al. (2004) detected the expression of 1α-hydroxylase in a β cell line and rat islets, but the physiological significance of vitamin D signalling in β cells during pancreas development is still unclear [12]. The pancreas is a target organ during “fetal programming”, and studies have already shown that maternal malnutrition affects the development of the fetal pancreas, causing reduced growth, in addition to dysfunction of the pancreatic islets and β cells [13]. On the other hand, the alterations caused by maternal VDD have not been extensively explored. A single study using maternal VDD in mice showed that male offspring exhibit a reduction in the expression of *Pdx1*, an important transcription factor for β cell development and function, indicating the importance of vitamin D in pancreas development and function [14]. However, little is known about the dimorphic effects of maternal VDD on offspring outcomes. Recent studies from our laboratory have demonstrated that maternal VDD selectively affects the development of type II muscle fibres in male but not female offspring rats [15]. The purpose of the present study was to examine the impact of maternal VDD, with or without Ca^2+^ supplementation, during pregnancy and lactation on the morphology and function of the endocrine pancreas, particularly with regard to insulin secretion, and the expression of genes involved in β cell development in male and female rat offspring in adulthood.

## 2. Results

### 2.1. Metabolites, Hormones, and Growth in Dams and Offspring

Maternal calcidiol status was measured at the end of lactation to confirm the efficacy of the diet. As expected, dams fed the VDD (11 ± 1.5 ng/mL) and VDD + Ca (10.4 ± 0.7 ng/mL) diets exhibited severe deficiency compared to those in the Ctrl group (33.8 ± 1.3 ng/mL; *p* < 0.05). In addition, both the VDD and VDD + Ca groups showed an increase in blood glucose levels (119 ± 3 and 121 ± 3, respectively, vs. 107 ± 2 mg/dL in the Ctrl group (*p* < 0.05)), indicating that VDD affects carbohydrate homeostasis in dams. VDD mothers also showed an increase in plasma corticosterone levels (18.1 ± 0.9 µg/dL) compared to those of controls (13.5 ± 1.7 µg/dL) (*p* < 0.05), but did not alter food intake (20 ± 3; 18 ± 2; 19 ± 1 g/24 h in Ctrl, VDD and VDD + Ca group, respectively). 

With respect to the offspring, at weaning (21 days old), VDD animals of both sexes presented hypovitaminosis D and hypocalcemia, whereas Ca^2+^ supplementation (VDD + Ca) in dams effectively normalized the Ca^2+^ concentration (Table 1). After weaning, the Ctrl and VDD offspring received a standard diet for up to 90 days. At this age, both males and females from the VDD groups showed normal levels of calcidiol, calcium, phosphorus, corticosterone, and glucagon (Table 2). During the first and second months of life, both the M-VDD and M-VDD + Ca groups gained less weight than the M-Ctrl group. At 90 days, M-VDD + Ca still had less weight gain than M-Ctrl, whereas M-VDD had recovered their body weight and remained the same as M-Ctrl (Figure 1A). F-VDD had lower body weight only in the second month of life, with weight returning to normal levels by 90 days. F-VDD + Ca also gained less weight in the first month, but their weight was re-established from the second month onwards (Figure 1B). At 90 days, no change in muscles and adipose tissue weight was observed in the VDD groups as compared to their respective controls (Table 2).

Taken together, these results show the efficacy of the model and that maternal VDD induces a delay in the growth rate of male and female offspring, mainly during the first 2 months of life, with these effects reversing at 90 days of age.

### 2.2. Glycemic Homeostasis and Insulinemia

As shown in Figure 2A,B, there were no differences in glycemic levels among groups of either sex during the experimental period. However, at weaning, the VDD male offspring (from the M-VDD and M-VDD + Ca groups) presented hypoinsulinemia compared to the controls (Table 1). This effect remained at 90 days of life (Figure 2C) but was not observed in fed females (Figure 2C) from the same groups or in deficient males and females under fasting conditions (Figure 2D). To investigate whether the basal hypoinsulinemia detected in the group of VDD male animals was related to alterations in the systemic metabolism of glucose, the glucose tolerance test (GTT) was performed, and the indicators of insulin resistance (HOMA-IR) and β cell secretion (HOMA-β) were calculated. The areas under the glycemic (Figure 3C) and insulinemic (Figure 3F) curves in response to glucose overload in the M-VDD group were not different from those in the M-Ctrl group. However, the M-VDD group presented greater glycaemia than the M-Ctrl group after 15 min of glucose administration (Figure 3A). At this time, the M-VDD+Ca group exhibited greater insulin secretion than the other groups (Figure 3D). Compared with males, VDD females had a larger area under the glycemic curve (Figure 3B,C), indicating greater glucose intolerance, but did not show changes in the insulin curve (Figure 3E,F). Like males, females in the VDD + Ca group showed greater insulin secretion after 15 min of glucose administration (Figure 3E). In addition, clear differences between the sexes were observed. Females from all groups secreted less insulin than males after 15 and 30 min of glucose administration, with no differences in glycaemia.

The HOMA-IR test is a mathematical model that allows indirect assessment of insulin resistance and is calculated from the values of blood glucose and fasting insulin levels. We found no differences in the HOMA-IR values between any of the experimental groups (Figure 3G). No change was observed in the constant rate for either the disappearance of glucose (Kitt), which was calculated from the curve obtained in the insulin tolerance test in males (0.86 ± 0.06 and 0.80 ± 0.07%/min in Ctrl and VDD, respectively) and females (1.08 ± 0.08 and 0.98 ± 0.1%/min in Ctrl and VDD, respectively). The HOMA-β was calculated to indirectly evaluate the function of β cells, and we found that both VDD and VDD + Ca males, but not females, exhibited a significant reduction in this index (Figure 3H), suggesting functional impairment of β cells. This difference was also observed between Ctrl females and Ctrl males. Collectively, these data suggest that maternal VDD impairs insulin secretion by β cells in male adult offspring without affecting systemic glucose metabolism. Although VDD females appear to be more glucose intolerant and secrete less insulin in response to glucose than males, their pancreas does not appear to be damaged by maternal VDD.

### 2.3. Pancreatic Islet Secretory Function

To further investigate how pancreatic islets from VDD animals respond to different glucose concentrations and to isolate potential effects of counterregulatory hormones, pancreatic islets were isolated during the GTT, and insulin secretion was determined. As shown in Figure 4A, the M-VDD group exhibited lower insulin secretion than the control group at physiological (5.6 mM) and high (11.1 mM) glucose concentrations. Similar results were found in the M-VDD+Ca group. No differences in the insulin secretory response to the highest glucose concentration (16.7 mM) were observed between the groups. Unlike males, F-VDD and F-VDD+Ca did not exhibit changes in insulin secretion in response to any glucose concentration tested (Figure 4B). In addition, the islets from females in all groups exhibited lower insulin secretion in response to glucose than did the islets from males, indicating clear sexual dimorphism. These data clearly demonstrate that maternal VDD impairs insulin secretion from pancreatic islets isolated from adult male offspring, whereas female offspring are fully protected from these deleterious effects.

### 2.4. Pancreatic Islet and β/α Cell Morphology

To investigate whether functional changes in islets were accompanied by structural changes, we analysed the morphology of pancreatic islets by H&E staining and of β/α cells by immunofluorescence. Maternal VDD led to a reduction in the total number of pancreatic islets in both males and females (Figure 5B). The total area of the pancreatic islets was smaller only in the M-VDD group (Figure 5C). This effect was associated with a decrease in islet mass when corrected for pancreas weight (Figure 5D) and animal weight (Figure 5E). To determine whether these pancreatic islet changes were associated with changes in β cell mass, we performed immunofluorescence anti-insulin staining and observed a significant decrease in absolute β cell mass in M-VDD islets but not in females (Figure 6B,C).

No morphological alterations were observed in islets or β cells in the F-VDD and F-VDD + Ca groups. Regarding the dimorphic differences, we observed that the Ctrl females had a smaller area and mass of pancreatic islets and β cells than did the males. Furthermore, no difference in alpha cell mass was detected, indicating that maternal VDD selectively affects pancreatic β cells in males (Figure 6D,E).

### 2.5. Expression of Genes Related to the Development and Function of β Cells

To clarify the molecular mechanisms involved in the morphofunctional changes induced by maternal VDD in male offspring, the expression of genes related to the development and function of β cells was investigated. As shown in Figure 7, *Ins1*, *Ins2*, *Pdx1*, and *SLC2A2* mRNA levels were significantly reduced in the pancreatic islets of the M-VDD and M-VDD + Ca groups, but this effect was not observed in the VDD females. In line with the other analysed parameters, females had lower *Pdx1* and *SLC2A2* gene expression than control males (Figure 7).

### 2.6. Methylation Analysis by Methylation-Sensitive Enzymatic Restriction (MSER) Associated with RT-qPCR

To investigate the mechanisms underlying the decreased expression of *Pdx1*, a transcription factor crucial for β cell identity, we conducted a methylation profile analysis of the promoter region of the *Pdx1* gene. According to the Ctrl group, which presented a hypomethylation profile, no difference was found in the methylation profiles of the VDD and VDD + Ca groups (Table 3).

## 3. Discussion

This study is the first to demonstrate that maternal VDD during pregnancy and lactation impairs β cell morphology and function in the offspring of male rats, whereas female rats are fully protected from these deleterious effects. Furthermore, these changes were not sufficient to alter glucose homeostasis at the age studied.

In male offspring, VDD induced hypoinsulinemia at the end of weaning, an effect that was accompanied by reduced plasma levels of calcidiol and hypocalcemia. This effect was also observed in animals of mothers that were supplemented with Ca^2+^, and metabolic effects (low calcium) were prevented, suggesting that the low levels of insulin were a direct consequence of maternal VDD during pregnancy and/or lactation. Given that hypoinsulinemia was also observed in adult life (90 postnatal days), a stage of life where the plasma levels of calcidiol had already been restored by a balanced diet, these findings reinforce the idea that the pancreas was permanently damaged by VDD during early life. These findings complement and extend a prior study that assessed the effect of maternal VDD on pancreatic islets in the adult offspring of mice [14], even though these authors have not ruled out the effects of hypovitaminosis from hypocalcemia.

The decrease in insulin found in the adult male offspring occurred without any change in the plasma levels of glucose or glucose tolerance, as estimated by the GTT. However, when evaluating in vitro insulin secretion, we found that pancreatic islets from M-VDD animals secreted less insulin than did those from controls under baseline conditions in vivo. These data are in agreement with several studies that have investigated the effects of vitamin D on the pancreas and its influence on insulin secretion. Vitamin D has been shown to increase insulin secretion through the VDR in both in vivo and in vitro studies [16]. Indeed, vitamin D supplementation has been demonstrated to exert beneficial effects on glycemic levels, glucose tolerance, and insulin secretion in experimental models of diabetes and in humans [10]. The fact that glucose metabolism was not affected in the VDD male offspring cannot be explained in the present study but could be related to the age of the animals. Indeed, young pups (6 weeks to 3 months) from mothers exposed to a low-protein diet showed better glucose tolerance when plasma insulin concentrations were reduced [16]. However, with advancing age, this effect reversed. At 15 months, male offspring showed increased glucose intolerance, and this condition was accompanied by increased insulin resistance [17].

The functional damage induced by maternal VDD in the islets from male offspring was associated with a drastic decrease in pancreatic islet area and β cell mass and was probably due to a failure in the development of the endocrine pancreas. In agreement with this notion, M-VDD animals showed a significant reduction in HOMA-β, an index of impairment of β cells. These data fit well with previous observations that the restriction of maternal nutrients reduces β cell mass and cell proliferation in neonates [18], an effect that is associated with reduced insulin secretion [3]. *Pdx1* is a transcription factor that is involved mainly in the formation of β cells [19], and a decrease in its expression leads to functional alterations in pancreatic insulin secretion and the development of type 2 diabetes in humans [19]. In previous studies, it was demonstrated that the protein content of Pdx1 is reduced in the pancreas of male mice whose mothers were fed a caloric restriction [20] or a VDD diet [14] during pregnancy. In agreement with these studies, we observed that *Pdx1*, *SCL2A2*, and *Ins* gene expression was reduced in the M-VDD group. It is well established that epigenetic changes, including acetylation and DNA methylation, are the main mechanisms involved in the repression of genes found in experimental models of fetal programming [21]. Accordingly, a study with intrauterine growth restriction showed that these male pups have increased *Pdx1* methylation [22]. Therefore, we evaluated the percentage of *Pdx1* methylation using the MESR technique. The VDD and VDD + Ca groups showed hypomethylation, as did the control groups. Consistent with our results, a study using a maternal malnutrition model revealed no change in the *Pdx1* methylation profile [23]. However, we cannot rule out other possible epigenetic mechanisms regulating *Pdx1* expression, such as histone modification.

Another well-established factor that can negatively affect the development of β cells is glucocorticoids. In vitro experiments using brotopancreatic preparations have shown that dexamethasone decreases the number of precursor cells that exclusively express *Pdx1* [24]. Indeed, excessive exposure of the fetus to maternal glucocorticoids decreases β cell mass and leads to low birth weight and, subsequently, glucose intolerance and hypertension [25]. In mice, maternal VDD induces a reduction in the placental expression of the gene (Hsd11b2) that codes for Hydroxysteroid 11-β Dehydrogenase 2, an enzyme that metabolizes glucocorticoids in the placenta, leading to an increase in the placental transfer of glucocorticoids to the fetus [26]. Interestingly, this enzymatic deficiency was observed only in the male fetus, but not in the female fetus [26]. Therefore, the present finding that mothers with VDD have elevated levels of corticosterone offers a potential explanation for the failure of pancreas development exclusively observed in male offspring.

With respect to the differences between males and females, we found that control females had a smaller islet area and β cell mass as well as lower expression of *Pdx1* and the other analysed genes than males. This could explain why normal females secrete less insulin in vitro and have lower basal insulin levels. Furthermore, one of the most interesting findings of this work is the sexual dimorphism found in the VDD offspring groups. Unlike males, females seem to be protected from the deleterious effects caused by maternal VDD in the endocrine pancreas. This finding is in accordance with a recent study from our laboratory showing that VDD during pregnancy and lactation induces morphological and functional changes in the skeletal muscle of male but not female offspring [15]. This difference may be related, in part, to female hormones, since estrogen has important protective effects on the pancreas and other tissues [27,28]. However, the importance of the placenta cannot be excluded, as the female placenta has been shown to adapt better to suboptimal conditions [29]. In addition, a clear sexual dimorphism in placental vitamin D metabolism has been demonstrated in trophoblast culture, where testosterone produced by the male fetus decreases the bioavailability of vitamin D [30]. Future studies employing castrated animals in different stages of life can help elucidate the VDD-induced sexual dimorphic response in the endocrine pancreas.

In summary, our study demonstrated that maternal VDD impairs the morphology and function of the endocrine pancreas in male offspring. The intriguing finding that females are fully protected from the damage induced by VDD clearly shows that sexual dimorphism is an important differentiating factor in the development of the pancreas in response to vitamin D and opens new avenues to investigate the underlying mechanistic differences between males and females.

## 4. Materials and Methods

### 4.1. Animals and Diets

The experimental procedures used in this study were performed according to the Brazilian College of Animal Experimentation and approved by Ribeirão Preto Medical School of the University of São Paulo-The Ethics Committee on Animal Use (CEUA 1241/2023).

Forty 5-week-old female Wistar Hannover rats were randomly assigned to a control diet (Ctrl; AIN93G with 000 IU Vitamin D3 per kg diet), a VDD diet (AIN93G without Vitamin D3 per kg diet), or a VDD diet supplemented with calcium (AIN93G without Vitamin D3 + Ca + P) for 6 weeks before conception and after conception (pregnancy and lactation) ad libitum. The diets were produced and marketed by Prag Solutions (Jaú, SP, Brazil), and their components are shown in Appendix A, Table A1. All animals were housed in a room with a 12 h light–dark cycle. After six weeks, the females were mated overnight with males of the same age who received only a control diet to form the experimental groups. Figure 8 shows the formation of groups. The administration of the diets continued until the end of lactation. The number of offspring was reduced to eight pups per mother. At weaning, male and female offspring were separated into three groups according to the mother’s diet: male and female offspring control (M-Ctrl and F-Ctrl, respectively), male and female offspring VDD (M-VDD and F-VDD, respectively), and male and female offspring VDD+Ca (M-VDD + Ca and F-VDD + Ca, respectively). Offspring received a Nuvilab standard diet (Nuvital, Quimtia) until 90 days of age and were weighed initially starting at 2 days of age each week until the first month and thereafter each month until study termination (90 days), at which time point tissues were harvested for analysis (Figure 8).

### 4.2. Glucose Tolerance Test and Insulin Tolerance Test

A glucose tolerance test (GTT) was performed on the offspring 1 week before euthanasia. Animals fasted for 8 h received a glucose solution (2 g/kg) by intraperitoneal injection. Blood samples were collected through an incision at the tip of the tail before glucose administration and 15, 30, 60, and 120 min after glucose administration to measure blood glucose and insulin levels (Accu-Chek Performa, Roche Applied Science, São Paulo, Brazil). Differences in group GTT curves were analysed by calculating total glucose and insulin “area under the curve” (AUC) (GraphPad Prism 6.05, GraphPad Software, La Jolla, CA, USA). Insulin was measured as described in the following section.

One week before the animals reached 3 months of age, another group of animals was fasted for 3 h. Thereafter, glucose was measured at time 0. Human recombinant insulin equivalent to 1 IU/kg body weight was then injected intraperitoneally. Other samples were collected at 5, 10, 15, and 30 min for blood glucose measurement. The constant rate for glucose disappearance (Kitt) was calculated from the slope of the regression line obtained with log-transformed glucose values between 0 and 30 min after insulin administration.

### 4.3. Dosage of Metabolites and Hormones

The serum calcidiol levels of the dams and their offspring were measured using a chemiluminescence analyser (DiaSorin, Liaizon^®^ XL, Austin, TX, USA). Serum insulin and glucagon concentrations were determined using ELISA (Merck Millipore, Burlington, MA, USA). Enzymatic kits from LabTest (Minas Gerais, Brazil) were used to determine the serum calcium and phosphorus concentrations. Serum corticosterone levels were measured using a specific radioimmunoassay as previously described [31].

### 4.4. Pancreatic Islets

Islets were isolated (collagenase method, n = 5/group), as described elsewhere [32]. Briefly, the pancreas was cannulated and inflated with cold Hanks’ solution (supplemented with 1 mg/mL materfetal bovine serum) containing 0.8 mg/mL collagenase IV (C5138, Sigma-Aldrich, Burlington, MA, USA). Then, the pancreas was removed and incubated in a 37 °C water bath for 20 min to allow digestion of the exocrine tissue. Subsequently, the tubes were vigorously shaken for approximately 15 s. The collagenase digestion was terminated by the addition of cold Hanks’ solution. The islets were manually collected under a stereomicroscope with a Pasteur pipette.

### 4.5. Insulin Content and Static Secretion

After the pancreatic islets were isolated, a static insulin secretion test was performed. For this, groups of 5 islets were first incubated for 1 h at 37 °C in a Krebs-bicarbonate buffer solution of the following composition (in mM): 115 NaCl, 5 KCl, 2.56 CaCl_2_, 1 MgCl_2_, and 5.6 glucose, supplemented with 0.5% bovine serum albumin and equilibrated with a mixture of 95% O_2_ and 5% carbon dioxide, pH 7.4. The medium was then replaced with fresh buffer containing 2.8, 5.6, 11.1, or 16.7 mM glucose and subsequently incubated for 1 h. At the end of the incubation, the medium was collected and stored at −80 °C for subsequent measurement of insulin content using ELISA. The insulin content of the islets was determined after extraction in an acid–ethanol solution (12 mM HCl in 70% ethanol). The islets were sonicated for 15 s, extracted overnight at 4 °C, and centrifuged for 10 min at 3000× *g*, after which the supernatant was frozen for insulin content analysis using ELISA.

### 4.6. Histology and Microscopy

To study the morphological and morphometric aspects of the endocrine pancreas, the pancreas (splenic fragment) was excised from 6 to 8 animals per group, fixed in neutral buffered formalin (pH 7.2), and kept for up to 48 h at room temperature. Afterwards, dehydration was performed with increasing concentrations of alcohol, and the fragments were cleared in xylol until they were embedded in paraffin. Subsequently, samples were sectioned at 5 μm in thickness in a microtome (Rotating Microtome Leica RM2255, Leica Byosystens, Wetzlar, Germany), incubated in a water bath, and stained according to the haematoxylin–eosin technique.

For β cell or α cell mass analyses, immunofluorescence labelling was performed as described above [33]. Briefly, antigenic retrieval buffer (citrate buffer) was applied under three cycles of microwave heating (low power) followed by incubation in phosphate buffer solution containing 0.2% Tween 20 (PBST)/3% bovine albumin for 1 h at room temperature. After washing in PBS, the sections were incubated for 2 h in a refrigerator with the following antibodies: insulin antibody (1:500; C27C9, Cell Signalling, Waltham, MA, USA and glucagon antibody (1:300; sc-514592, Santa Cruz Biotechnology, Dallas, TX, USA) in PBST/3% bovine serum albumin. After being washed in PBS, the sections were incubated for 1 h with specific secondary antibodies at room temperature, namely, Alexa Fluor 488-conjugated rabbit and Alexa Fluor 555-conjugated rabbit antibodies (1:300–1:500) in PBST/3% bovine serum albumin. After the final washes in PBS, the sections were mounted with Vectashield antifading mounting medium supplemented with DAPI (cat. H-200, Vector Laboratories, Newark, CA, USA).

Both haematoxylin–eosin and immunofluorescence imaging sections were then fully scanned using an Olympus BX61VS (Olympus Corporation, Tokyo, Japan) fluorescence microscope, and morphometric analyses were performed using ImageJ software (Fiji is Just; version 1.52p; National Institutes of Health). Each section of the pancreas was systematically evaluated in terms of the total section area (pixels) and total insulin- or glucagon-positive areas (pixels). The relative mass of β or α cells (expressed in %) was calculated by dividing the sum of the insulin- or glucagon-positive areas in the entire area of the pancreas × 100. The absolute mass of β or α cells (expressed in mg) was calculated by multiplying the relative mass of β or α cells (%) by the mass of the pancreas (whole pancreas, not just the splenic fraction).

### 4.7. Quantitative PCR

Gene expression levels were detected by using reverse transcription polymerase chain reaction. Total RNA was extracted from pancreatic islets using the TRIzol (Invitrogen, Waltham, MA, USA) method and then transcribed into cDNA using the SuperScript IV First-Strand Synthesis System (Invitrogen) according to the manufacturer’s instructions. Quantitative PCR was performed using PowerUp SYBR Green Master Mix (Thermo Fisher Scientific, Waltham, MA, USA) with the primers detailed in Table 4, and the results were normalized to those for *β-actin*.

### 4.8. Genomic DNA Extraction

Methylation levels were measured using the enzyme restriction method. Pancreatic islets were homogenized in Tissue Lyzer in PBS buffer (500 µL) and lysis buffer (450 µL) containing 0.32 M sucrose, 12 mM TrisHCl, 5 mM MgCl_2_, and 1% Triton 100×. After centrifugation at 13,000 rpm for 20 s, the samples were digested with proteinase K (buffer containing 0.375 M NaCl, 12 M EDTA, Milli-Q H_2_O, 20% SDS, and proteinase K) at 55 °C overnight. NaCl was added to precipitate the denatured proteins. This process was followed by washing with absolute ethanol and 70% ethanol. The pellet was resuspended in 50 µL of Milli-Q water, incubated at 37 °C for 1 h, and stored at −20 °C. Quantification of genomic DNA was performed on a Nanodrop (Thermo Fisher Scientific, Waltham, MA, USA) the following day.

### 4.9. Methylation-Sensitive Enzymatic Restriction Associated with RT-qPCR

DNA methylation levels were measured using real-time PCR associated with enzymatic restriction with methylation-sensitive HpaII [34]. For enzymatic digestion, 2 µL of genomic DNA (50 ng/µL) was used in three separate solutions: (1) control without enzymatic digestion; (2) solution containing the methylation-sensitive HpaII enzyme (ER0512, Thermo Fisher); and (3) solution containing MspI enzyme (ER0541, Thermo Fisher) as a digestion control. The amount of product after digestion was quantified using the StepOne Plus System (Applied Biosystems, Waltham, MA, USA) with PCR parameters and melting curves suitable for product verification. The PowerUp™ SYBR™ Green Master Mix (Thermo Fisher Scientific, Waltham, MA, USA) was used with 10 pmol of specific primers for the *Pdx1* promoter region (forward: 5′ TAAGGCAGGGCCAGGCCAATGGTG 3′, reverse: 5′ GGAGCTACAA-GCCAGGCCTTAAGGC 3′). The percentage of methylated sample was calculated using (1/2)^n^, where n is the number of cycles, which was obtained by subtracting the average Ct of digested DNA with HpaII enzyme from the average Ct of undigested DNA [34].

### 4.10. Statistics

The data are presented as the mean ± standard error of the mean. Statistical significance was assessed using two-way analysis of variance followed by a Tukey post hoc test for multiple comparisons or Student’s t test for the means from different groups. *p* ≤ 0.05 was considered significant.

## Figures and Tables

**Figure 1 ijms-25-04136-f001:**
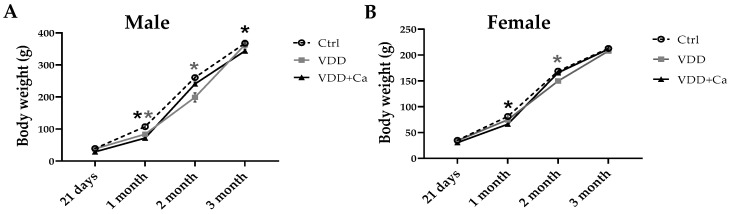
Effect of maternal vitamin D deficiency (VDD) alone or with calcium supplementation (VDD + Ca) on the growth curves of male (**A**) and female (**B**) offspring from weaning (21 days) until 3 months of life. The data are expressed as the mean ± SEM (n = 8). Statistical significance was determined at *p* < 0.05 * vs. control diet.

**Figure 2 ijms-25-04136-f002:**
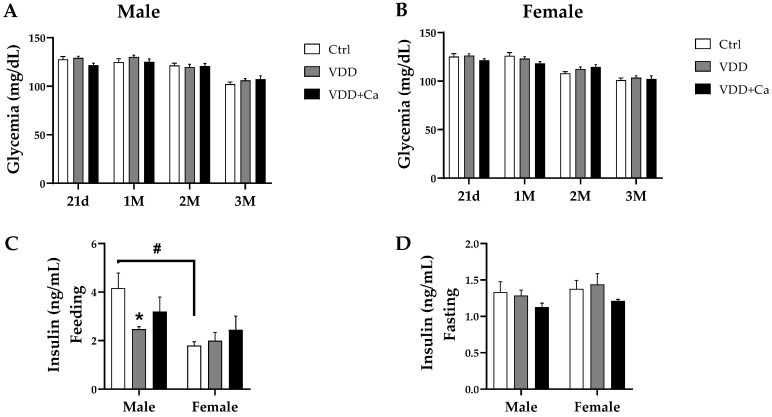
Effect of maternal vitamin D deficiency (VDD) alone or with calcium supplementation (VDD + Ca) on plasma glucose (**A**,**B**) and insulin levels under fed (**C**) and fasted (**D**) conditions in male (M) and female (F) rat offspring. The data are expressed as the mean ± SEM (n = 8). Statistical significance was determined at *p* < 0.05 * vs. the Ctrl diet; # vs. the male group.

**Figure 3 ijms-25-04136-f003:**
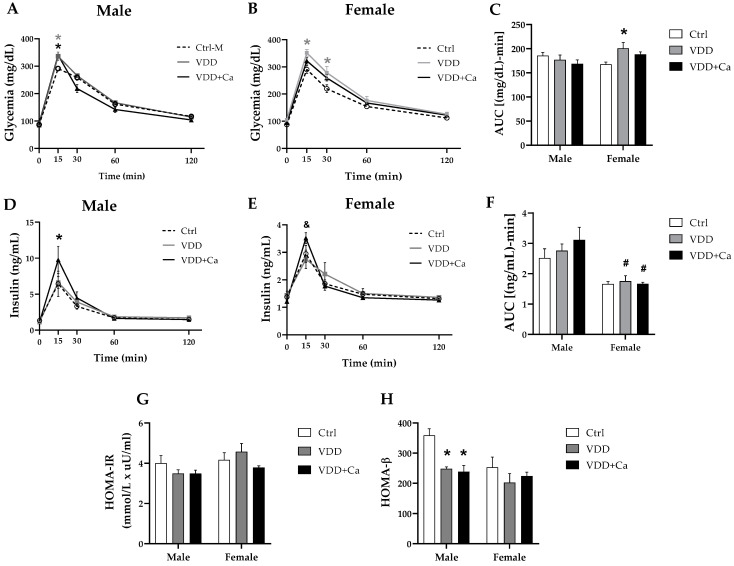
Effect of maternal vitamin D deficiency (VDD) alone or with calcium supplementation (VDD + Ca) on the oral glucose tolerance test (2 g/kg) in adult male (M) and female (F) offspring. Glycemic curves (**A**,**B**) and the area under the glycemic curve (**C**) were plotted along with insulinemia curves (**D**,**E**) and the area under the insulinemia curve (**F**) for each GTT. Insulin resistance homeostasis model (HOMA-IR) (**G**) and steady-state β cell function (HOMA-β) (**H**) were evaluated using blood glucose values and fasting insulinemia. The data are expressed as the mean ± SEM (n = 6). Statistical significance was determined at *p* < 0.05 * vs. Ctrl diet; # vs. male group; & vs. VDD group.

**Figure 4 ijms-25-04136-f004:**
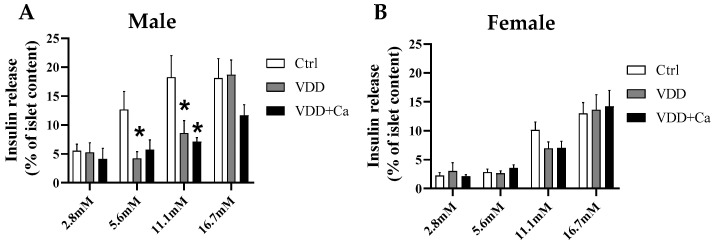
Effect of maternal vitamin D deficiency (VDD) alone or supplemented with calcium (VDD + Ca) on glucose-stimulated insulin secretion from pancreatic islets isolated from adult male (**A**) and female (**B**) offspring. Following isolation, the pancreatic islets were incubated for 1 h in Hank’s solution containing various glucose concentrations: 2.8 mM, 5.6 mM, 11.1 mM, and 16.7 mM. The results are presented as the mean ± SEM. Statistical significance was determined at *p* < 0.05, with * vs. the Ctrl diet.

**Figure 5 ijms-25-04136-f005:**
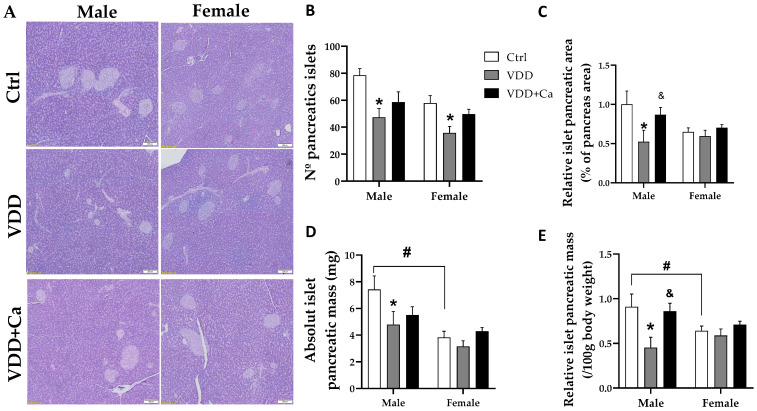
Effect of maternal vitamin D deficiency (VDD) alone or supplemented with calcium (VDD + Ca) on the pancreatic islet mass of adult male (M) and female (F) rat offspring. (**A**) Representative images of pancreatic sections stained with H&E. Scale bars indicate 200 μm at a final magnification of 200×. The number of pancreatic islets (**B**), the relative area of the pancreatic islets (**C**), and the respective absolute and relative masses (**D**,**E**) were measured. The data are expressed as the mean ± SEM (n = 8). Statistical significance was determined at *p* < 0.05 * vs. the Ctrl diet; # vs. the male group; and & vs. the VDD + Ca group.

**Figure 6 ijms-25-04136-f006:**
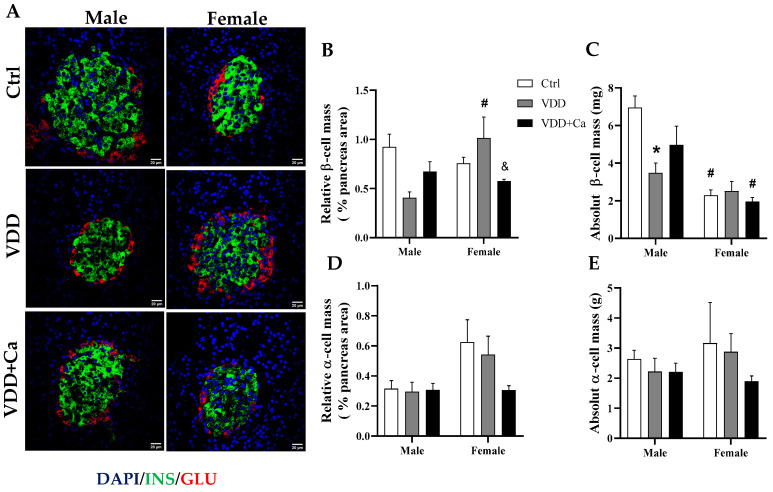
Effect of maternal vitamin D deficiency (VDD) alone or supplemented with calcium (VDD + Ca) on pancreatic β- and α-cell mass in adult male (M) and female (F) rat offspring. Left, (**A**) panel with representative images of pancreatic sections double-immunostained with insulin and glucagon. The scale bars indicate 200 μm at a final magnitude of 200×. The relative β- (**B**) and α-cell masses (**D**) and the respective absolute masses of β and α cells (**C**,**E**). The data are expressed as the mean ± SEM (n = 8). Statistical significance was determined at *p* < 0.05 * vs. the Ctrl diet; # vs. the male group; and & vs. the VDD group.

**Figure 7 ijms-25-04136-f007:**
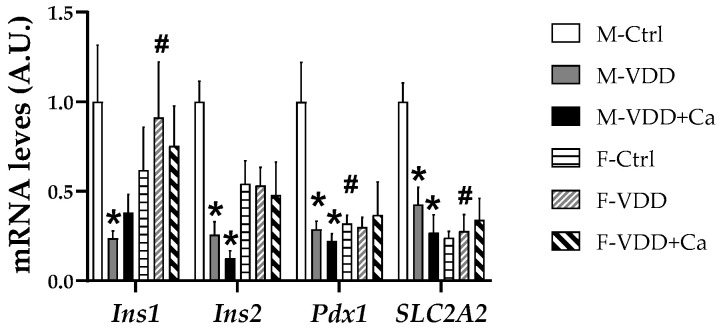
Effect of maternal vitamin D deficiency (VDD) alone or supplemented with calcium (VDD + Ca) on the gene expression of insulin (*Ins1* and *Ins2*), *Pdx1*, and *SLC2A2* in pancreatic islets from adult male (M) and female (F) rat offspring. The data are expressed as the mean ± SEM (n = 6). Statistical significance was determined at *p* < 0.05 * vs. the Ctrl diet group; # vs. the male group.

**Figure 8 ijms-25-04136-f008:**
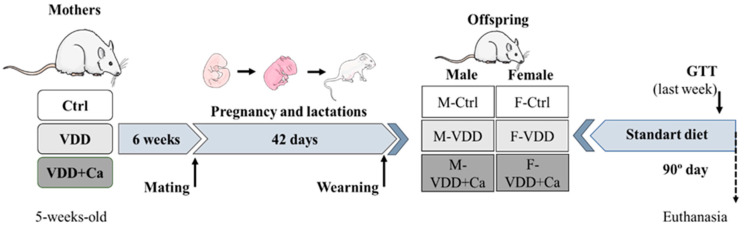
Experimental protocol. The female rats were randomly separated into three groups and fed a control diet (Ctrl; AIN93G-1000 IU Vitamin D3/kg diet), vitamin D deficient (VDD) diet (AIN93G without Vitamin D3), or a vitamin D deficient diet supplemented with calcium (VDD + Ca) (AIN93G without Vitamin D3 + Ca + P) for 6 weeks and then allowed to mate with male rats. After weaning, the pups were separated according to sex (M: male and F: female) and maternal diet. Parts of the figure were made using Medical Art. Medical Art by Servier is licenced under a Creative Commons Attribution 3.0 Unported Licence (https://creativecommons.org/licences/by/3.0/, accessed on 5 April 2024).

**Table 1 ijms-25-04136-t001:** Effect of maternal vitamin D deficiency (VDD) alone or with calcium supplementation (VDD + Ca) on different parameters of mineral metabolism and plasma insulin in male (M) and female (F) offspring at weaning (21 days old).

	M-Ctrl	M-VDD	M-VDD + Ca	F-Ctrl	F-VDD	F-VDD + Ca
Calcidiol (ng/mL)	29 ± 0.9	5 ± 0.3 *	6 ± 0.5 *	28 ± 0.8	5 ± 0.4 *	5 ± 0.2 *
Calcium (mg/dL)	8 ± 0.6	6.8 ± 0.1 *	9.2 ± 0.2	8 ± 0.3	6.8 ± 0.2 *	8 ± 0.2
Insulin (ng/mL)	2 ± 0.1	0.8 ± 0.1 *	0.5 ± 0.1 *	1.3 ± 0.3	1.4 ± 0.3	1.1 ± 0.2

The data are expressed as the mean ± SEM. * *p* < 0.05 vs. the Ctrl group.

**Table 2 ijms-25-04136-t002:** Effect of maternal vitamin D deficiency (VDD) alone or with calcium supplementation (VDD + Ca) on hormones, metabolites, and tissue weight in adult male (M) and female (F) offspring.

	M-Ctrl	M-VDD	M-VDD + Ca	F-Ctrl	F-VDD	F-VDD + Ca
Calcidiol (ng/mL)	48.4 ± 1.5	43.0 ± 1.3	44.9 ± 1.4	48.1 ± 1.1	51.5 ± 1.1	53.6 ± 1.1
Calcium (mg/dL)	9.4 ± 0.5	9.4 ± 0.5	9.7 ± 0.6	9.4 ± 0.5	9.5 ± 0.8	9.5 ± 0.5
Phosphorus (mg/dL)	6.3 ± 0.9	6.3 ± 1.1	6.4 ± 1.0	5.3 ± 0.7	6.0 ± 0.9	4.9 ± 0.8
Corticosterone (µg/dL)	11.5 ± 1.7	13.3 ± 2.2	6.9 ± 2.1	20.5 ± 2.7	17.8 ± 2.7	18.3 ± 2.2
Glucagon (ng/mL)	0.26 ± 0.05	0.22 ± 0.06	0.33 ± 0.05	0.26 ± 0.04	0.27 ± 0.03	0.25 ± 0.04
Retroperitoneal WAT (g/100 g BW)	1.17 ± 0.19	1.06 ± 0.11	1.67 ± 0.11	0.70 ± 0.11	0.84 ± 0.09	0.71 ± 0.06 #
Epididymal WAT (g/100 g BW)	1.07 ± 0.12	0.95 ± 0.07	1.11 ± 0.05	-	-	-
Periovarian WAT (g/100 g BW)	-	-	-	0.4 ± 0.1	0.44 ± 0.03	0.51 ± 0.03
EDL (mg/100 g BW)	43 ± 1.3	46 ± 0.2	44 ± 0.9	46 ± 3.3	46 ± 2.4	45 ± 1.5
Soleus (mg/100 g BW)	36 ± 2.9	37 ± 1.5	35 ± 1.4	36 ± 3.3	42 ± 0.5	39 ± 1.2

The data are expressed as the mean ± SEM. # vs. male group.

**Table 3 ijms-25-04136-t003:** Methylation analysis by MESR associated with RT-qPCR.

		Average CT	n	Met%
**Male**	U Ctrl	27.2 ± 0.6		
	D Ctrl	32.0 ± 1.1	4.8	0.04
	U VDD	26.8 ± 0.3		
	D VDD	32.8 ± 0.4	6	0.01
	U VDD + Ca	26.2 ± 1.1		
	D VDD + Ca	32.3 ± 1.3	6.1	0.01
**Female**	U Ctrl	26.7 ± 1.2		
	D Ctrl	32.4 ± 1.0	5.7	0.03
	U VDD	27.2 ± 1.2		
	D VDD	33.2 ± 0.8	6	0.03
	U VDD + Ca	26.1 ± 1.2		
	D VDD + Ca	32 ± 1	5.9	0.02

U, undigested DNA; D, DNA digested by HpaII; Average ct, average of threshold cycles; n, number of cycles obtained by subtraction of average ct of digested DNA by average CT of undigested DNA; Met%, percentage of methylated DNA obtained by (1/2)^n^.

**Table 4 ijms-25-04136-t004:** Oligonucleotide primers used for qPCR analysis.

Gene	Reverse	Reverse
Rat *Ins1*	CCATCAGCAAGCAGGTCAT	TGTGTAGAAGAAACCACGTTCC
Rat *Ins2*	TCTTCTACACACCCATGTCCC	GGTGCAGCACTGATCCAC
Rat *Pdx1*	CCGCGTTCATCTCCCTTT	TGCCCACTGGCTTTTCCA
Rat *SLC2A2*	GTCCAGAAAGCCCCAGATACC	TGCCCCTTAGTCTTTTCAAGCT
Rat *β-actin*	TGAAGTGTGACGTTGACATCC	ACAGTGAGGCCAGGATAGAGC

## Data Availability

The original contributions presented in the study are included in the article, further inquiries can be directed to the corresponding authors.

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
