# Peer review of "Maternal Vitamin D Deficiency Impairs the Development of β Cells in Offspring Rats in a Sex-Dependent Manner"

_ijms, 2024, doi:10.3390/ijms25084136_

Round 1

Reviewer 1 Report

Comments and Suggestions for Authors

The article "Maternal Vitamin D deficiency impairs the development of β cells in offspring rats in a sex-dependent manner" provides significant insights into the consequences of maternal Vitamin D deficiency on the endocrine pancreatic development and insulin secretion in rat offspring.

    Relevance of the Topic: the topic is highly relevant as Vitamin D deficiency (VDD) is a common issue globally and its implications on offspring health are of significant concern. The study's focus on the sex-dependent impact of maternal VDD on β cell development and insulin secretion extends our understanding of fetal programming and metabolic diseases, underlining its relevance.

    Quality of Writing: the article is well-structured, with clear sections delineating objectives, methods, results, and conclusions. The writing is coherent, facilitating comprehension of complex biological processes. Terminology and concepts are adequately explained, making the content accessible to readers with varying expertise levels.

    Current Relevance and Quality of Statistics: the statistics employed, such as metabolic and hormone measurements, and gene expression analysis, are pertinent and methodologically sound. They provide a robust quantitative foundation for the study's conclusions. The study's timely nature, considering the prevalent concern over VDD, ensures its findings are of current interest.

    Research Methods: the methodological framework is comprehensive, incorporating dietary manipulations, glucose tolerance tests, insulin level assessments, pancreatic islet analyses, and gene expression studies. The use of both in vivo and in vitro approaches strengthens the validity of the findings. However, a more detailed exploration of potential epigenetic mechanisms could enhance the depth of understanding.

    Discussion and Conclusions: the discussion adeptly contextualizes the findings within the broader scientific literature, noting the study's novelty and its alignment or divergence with previous work. The conclusion that maternal VDD has a sex-dependent impact on offspring, particularly affecting male β cell development, is well-supported by the data. The recognition of sexual dimorphism offers valuable insights for future research and potential therapeutic interventions.

Results Interpretation: while the discussion effectively links the results to existing literature, the authors could delve deeper into potential biological mechanisms and clinical significance of their findings. Such an analysis could enhance the study's contribution to the scientific community and practice.

Recommendations for Future Research: suggesting specific directions for future research based on the findings could improve the study's practical value. It would also aid in shaping the research agenda, especially regarding the study of Vitamin D effects across different populations and conditions.

In summary, the article provides a thorough and insightful examination of the effects of maternal Vitamin D deficiency on offspring pancreatic development, with a notable emphasis on sex-dependent outcomes. Its methodological rigor, comprehensive analysis, and relevance to public health concerns render it a valuable contribution to the field of endocrinology and developmental biology.

Comments on the Quality of English Language

While the language in the given response is clear and largely adheres to academic writing conventions, subtle nuances, fluency, and idiomatic expressions typical of native English writing can further improve its impact and readability. In summary, although the text is well-written, a review by a language expert could enhance its literary quality and ensure it aligns with the highest standards of academic communication.

Author Response

Dear reviewer, thank you for the review and comments. 

Following the reviewer’s suggestion, the text has been reviewed by a English language expert (please see the attached certificate).

We appreciate the careful review of the manuscript. The reviewer's questions, suggestions and corrections greatly improved the quality of the manuscript and we believe that it will now be acceptable for publication in IJMS.

Reviewer 2 Report

Comments and Suggestions for Authors

Major comments:

1. Please be mot specific, when you use the term "vitamin D". Do you mean "vitamin D3", "25(OH)D3" or "1,25(OH)2D3"? Moreover, please retain from the abbreviation "Vit.D" and write it out. Finally, please do not use two different term for the same compound (cholecalciferol, calcidiol, calcitriol etc)).

2. Please present all figures in a consistent way concerning font size, graph size etc.

Minor comments: 

1. Please define all abbreviations at their first time use them then consistently.

2. Have a space between numbers and units.

3. Gene name abbreviations should be in italic and have all gene name abbreviations based on latest nomenclature.

Comments on the Quality of English Language

Moderate improvements necessary.

Author Response

Dear reviewer, we thank you for your careful review of the manuscript. The reviewer's questions, suggestions and corrections greatly improved the quality of the manuscript and we believe that it will now be acceptable for publication in IJMS.

1. Please be mot specific, when you use the term "vitamin D". Do you mean "vitamin D3", "25(OH)D3" or "1,25(OH)2D3"? Moreover, please retain from the abbreviation "Vit.D" and write it out. Finally, please do not use two different term for the same compound (cholecalciferol, calcidiol, calcitriol etc)).

-To be more specific, we defined in the text that "vitamin D" means calcitriol (1,25 dihydroxyvitamin D) (Pg 1; line 44). Moreover, "Vitamin D" is not more abbreviated.

2. Please present all figures in a consistent way concerning font size, graph size etc.

-The figures were standardized in the same format.

Minor comments: 

1. Please define all abbreviations at their first time use them then consistently.

- It has been done

2. Have a space between numbers and units.

-It has been corrected

3. Gene name abbreviations should be in italic and have all gene name abbreviations based on latest nomenclature.

-Gene name abbreviations are now italicized and abbreviated according to the latest terminology.

Reviewer 3 Report

Comments and Suggestions for Authors

The authors investigated the impact of maternal VDD during pregnancy and lactation on the morphology and function of the endocrine pancreas in male and female rat offspring. The researchers found that maternal VDD impaired insulin secretion, reduced pancreatic islet area and β-cell mass, and decreased the expression of key genes related to β-cell development and function (Ins1, Ins2, PDX-1, GLUT-2) in adult male offspring. These effects were observed even when the offspring's vitamin D levels were restored to normal in adulthood, suggesting permanent damage to the pancreas caused by early life VDD exposure.

  1. The authors did not provide details on how they determined the sample size for each group.
  2. The study only examined the offspring at weaning and 90 days of age. Including additional time points, particularly during the early postnatal period and later in adulthood, could provide a more comprehensive understanding of the long-term effects of maternal VDD on pancreatic development and function.
  3. The authors did not include a pair-fed control group to account for potential differences in food intake between the control and VDD groups.
  4. While the study performed glucose tolerance tests and measured fasting insulin levels, incorporating additional tests such as insulin tolerance tests or hyperglycemic clamps could provide a more comprehensive assessment of glucose homeostasis and insulin sensitivity.
  5. Although the study identified sex-specific differences in the effects of maternal VDD, the underlying mechanisms were not thoroughly investigated.
  6. Given the importance of VDR and local vitamin D activation in pancreatic islets, measuring the expression levels of these components could help elucidate the direct effects of vitamin D on pancreatic development and function.
  7. While the study focused on β-cell morphology and function, investigating the effects of maternal VDD on α-cell mass and glucagon secretion could provide a more comprehensive understanding of the impact on the entire endocrine pancreas.
  8. Maternal VDD may affect offspring growth and body composition, which could indirectly influence pancreatic development and function. Measuring body weight, length, and body composition at different time points could help control for these potential confounding factors.

Author Response

Dear reviewer, thank you for your comments and review suggestions. After careful consideration of the criticisms and suggestions, all pertinent, we made several modifications to the original manuscript to meet the requests. Follow the responses below:

1.The authors did not provide details on how they determined the sample size for each group.

-The sample size was based on our previous experience with an animal model of vitamin D deficiency (J Cachexia Sarcopenia Muscle. 2022 Aug;13(4):2175-2187. doi: 10.1002/jcsm.12986) and was approved by The Ethics Committee on Animal Use from our Institution (CEUA 52/2018).

2. The study only examined the offspring at weaning and 90 days of age. Including additional time points, particularly during the early postnatal period and later in adulthood, could provide a more comprehensive understanding of the long-term effects of maternal VDD on pancreatic development and function.

-As shown in Table 1, at weaning (21 days old), the Vitamin D deficient male offspring (M-VDD and M-VDD+Ca groups) presented hypoinsulinemia compared to controls. Further studies should investigate the morphology and function of beta cells at this age and in adult offspring rats older than 90 days.

3. The authors did not include a pair-fed control group to account for potential differences in food intake between the control and VDD groups.

-We did not, but it is unlikely that this parameter may have affected the results since food intake was not altered by vitamin D deficiency (VDD) in dams. This information is now cited in the text in the new version of the Ms (Pg 2; line 88).

4. While the study performed glucose tolerance tests and measured fasting insulin levels, incorporating additional tests such as insulin tolerance tests or hyperglycemic clamps could provide a more comprehensive assessment of glucose homeostasis and insulin sensitivity.

-Following the reviewer’s suggestion, we added the insulin tolerance test (ITT) results. In agreement with the HOMA IR-test, the ITT was not different between the VDD and control groups (Pg 5;  line 212).

5. Although the study identified sex-specific differences in the effects of maternal VDD, the underlying mechanisms were not thoroughly investigated.

-We investigated epigenetic mechanisms because they could account for the sex-specific differences in the effects of maternal VDD. However, we did not find differences between the groups.

6. Given the importance of VDR and local vitamin D activation in pancreatic islets, measuring the expression levels of these components could help elucidate the direct effects of vitamin D on pancreatic development and function.

-We agree with the reviewer but, unfortunately, the primers we used to detect VDR expression in the pancreas did not work out, and we were unable to demonstrate it.

7. While the study focused on β-cell morphology and function, investigating the effects of maternal VDD on α-cell mass and glucagon secretion could provide a more comprehensive understanding of the impact on the entire endocrine pancreas.

-There was a misunderstanding by the reviewer. We investigated the effects of maternal VDD on α-cell mass. In contrast to beta-cells, α-cells were unaffected by VDD in male and female rats (Fig. 6). To reinforce this point, we have included in the new version of the Ms, the plasma levels of glucagon (see Table 2). (see Table 2).

8. Maternal VDD may affect offspring growth and body composition, which could indirectly influence pancreatic development and function. Measuring body weight, length, and body composition at different time points could help control for these potential confounding factors.

-We agree with the reviewer's comments about the importance of investigating body composition at different time points. Indeed, maternal VDD affected the offspring's growth during the 1st and 2nd months of life, but these effects were reverted in 3rd month (Fig. 1). Unfortunately, we did not measure the body composition and length at 1st and 2nd month. Still, we measured the mass of different muscles and adipose tissues at the moment of euthanasia (3rd month), and we did not find differences between groups. We have included these data in the new version of the Ms (see Table 2).

We thank the careful review of the manuscript. The questions, suggestions, and corrections by the reviewer have greatly improved the quality of the manuscript and we believe that it will now be acceptable for publication in IJMS.

Round 2

Reviewer 2 Report

Comments and Suggestions for Authors

1. Vitamin D nomenclature is still not correct and "Vit D" still not fully eliminated  from manuscript. Please realize that "vitamin D" is not always identical with 1,25(OHI2D3 when used in a sentence, since sometimes rather vitamin D3 or 25(OH)D3 is meant. Please revise the manuscript consistently.

2. It should be rather "Pdx1" and not "Pdx-1"

Comments on the Quality of English Language

Moderate corrections needed

Author Response

Dear Reviewer,
The required changes have been made and highlighted in yellow.

  1. Vitamin D nomenclature is still not correct and "Vit D" still not fully eliminated  from manuscript. Please realize that "vitamin D" is not always identical with 1,25(OHI2D3 when used in a sentence, since sometimes rather vitamin D3 or 25(OH)D3 is meant. Please revise the manuscript consistently.
    -To comply with the reviewer's request, we revised the manuscript, eliminated "Vit D," defined that vitamin D means calcitriol (Pag.1; Line 44) and vitamin D3 means cholecalciferol (Pag.2; Line.50). Moreover, we replaced calcitriol with vitamin D to make this point more straightforward.
    2. It should be rather "Pdx1" and not "Pdx-1"
    -This error has been corrected.
    3. The text had already been reviewed by an English language expert (please see the attached certificate).
